# Association between Familism and Mental Health in College Adolescents during the COVID-19 Pandemic

**DOI:** 10.3390/ijerph20054149

**Published:** 2023-02-25

**Authors:** Cecilia Mayorga-Muñoz, Leonor Riquelme-Segura, Elisa Delvecchio, Saulyn Lee-Maturana

**Affiliations:** 1Departamento de Trabajo Social, Facultad de Educación, Ciencias Sociales y Humanidades, Universidad de La Frontera, Temuco 4780000, Chile; 2Department of Philosophy, Social and Human Sciences and Education, University of Perugia, 06123 Perugia, Italy; 3Escuela de Psicología y Filosofía, Universidad de Tarapacá, Arica 1000000, Chile

**Keywords:** familism, allocentrism, idiocentrism, mental health, adolescents

## Abstract

Familism, also known in the literature as allocentrism, is the cultural propensity of a society to place the family at the center of its value system. Adherence to this value has been related to less depressive symptomatology in young people; however, these results are not conclusive, since it has also been found that the influence of familism on depressive symptoms is more indirect than direct. This study aimed to explore the direct relationships between familism (allocentrism and idiocentrism) and mental health (depression, anxiety, and stress). Methodologically, the study had a non-experimental, cross-sectional, descriptive, and correlational design. A sample of 451 Chilean university students responded to an instrument composed of the subscales allocentrism, idiocentrism, depression, anxiety, and stress during the COVID-19 pandemic. The results showed that family allocentrism was positively and significantly associated with depression (*γ* = 0.112, *p* < 0.05), anxiety (*γ* = 0.209, *p* < 0.001), and stress (*γ* = 0.212, *p* < 0.001), and family idiocentrism was negatively and significantly linked with depression (*γ* = −0.392, *p* < 0.001), anxiety (*γ* = −0.368, *p* < 0.001), and stress (*γ* = −0.408, *p* < 0.001). These findings contribute to supporting actions to reduce negative symptomatology and promote greater well-being in university students.

## 1. Introduction

Familism has been defined as the cultural disposition of a society to place the family at the center of its value system, prioritizing collective interests over personal interests, autonomy, and individualism [1]. Familism is a predominant value in Hispanic and Mediterranean cultures, where respect, support, cohesion, obligation, and obedience to the family are emphasized [2,3].

From a psychological standpoint, familism contributes to loyalty, emotional attachment, intergenerational solidarity, and economic support among the members of the group. These elements cause a high sense of belonging, which is expressed as frequent interactions that involve cultural behaviors [4,5]. Triandis [6] refers to this as personal collectivism or allocentrism. He also refers to the opposite behaviors as idiocentrism or personal individualism [7]. However, both allocentrism and idiocentrism have important implications on people’s emotional, cognitive, and motivational experiences [8]. Allocentrism and idiocentrism are considered part of familism.

Cohesive and nurturing family environments are associated with psychological and physical health. Greater familism promotes the prosocial behavior of young people [9], as well as self-esteem, satisfaction with life, a sense of a meaning, general well-being [10], and family satisfaction. Furthermore, it can stimulate commitment, cohesion, and support; and it can protect against depressive symptoms [11,12].

Coincidentally, Stein et al. [13] and Campos et al. [14] suggested that attachment to the family is a protective factor against depressive symptoms in adolescents. These authors also suggested that it can reduce the effects of stress, which is considered a risk factor for depression and physical illness [15,16]. Even more, it has been reported that emerging adults showing family attachment tend to have greater psychological adjustment [17].

However, these findings are not conclusive, since it has been found that the influence of familism on depressive symptoms is more indirect than direct [14]. For example, some studies have not been able to demonstrate a direct relationship between familism and depressive symptoms in university students [11,14]. Other studies have evaluated whether familism constitutes a protective factor when adolescents face stressful situations [13]. Some studies have shown more depressive symptoms with higher levels of familism, while others have not found any relationships and some have identified familism as a protective factor [2]. Similarly, Lee and Solheim [8] suggested that familism could contradict the need for individuation in young people as a part of their development and could generate psychological discomfort. Therefore, the evidence is still insufficient.

The literature has shown that familism changes according to age, varying in adherence during youth [13]. Additionally, differences by gender have been identified, showing female university students supporting the values of familism more than their male peers [11]. Hence, cultural gender roles could differentially determine the protective role of familism and its influence on the development of children [15,18].

Three criteria have been identified in which familism is manifested or expressed: *structural*, referring to the physical proximity between family members; *behavioral*, which refers to personal behavior in accordance with family values and expectations; and *attitudinal*, which includes thoughts and feelings toward the family [1,2,4]. These criteria have been used as theoretical base for the construction of instruments to measure familism. Instruments to measure familism consider both allocentrism and idiocentrism. However, there is little research studying the relationship between collectivism and individualism (or allocentrism and idiocentrism) and negative emotional symptoms, and its incidence on university students’ mental health.

Mental health is defined as a “well-being state in which the individual carries out his capabilities, overcomes the normal stresses of life, works productively and fruitfully, and contributes to their community” [19]. There is evidence of a decrease in the state of mental health, general health, and quality of life of the population during the COVID-19 pandemic [20]. Some studies have shown anxious and depressive symptoms in the population under 35 years of age [21], a prevalence of post-traumatic stress and post-confinement depressive symptoms in Chinese university students [22], and a greater presence of emotional problems in adolescents, with anxious and depressive symptoms being more frequent in women than in men [23,24,25,26].

Considering that mental health symptoms are mediated by contextual and cultural factors [27], it is interesting to explore the relationship between cultural variables such as familism and mental health in young Chileans during the COVID-19 pandemic. In this study, youth was understood as the age group between 15 and 24 years, a category that includes adolescents and emerging adults [23,28].

The evidence presented guides the following hypotheses:

**Hypothesis 1 (H1).** 
*Family allocentrism is positively associated with depression, anxiety, and stress.*


**Hypothesis 2 (H2).** 
*Family idiocentrism is negatively associated with depression, anxiety, and stress.*


Based on the above, this study aimed to explore the relationships between allocentrism and idiocentrism and depression, anxiety, and stress in Chilean university students during the COVID-19 pandemic.

## 2. Materials and Methods

The present investigation used a quantitative approach with a non-experimental, cross-sectional design and a descriptive correlational scope [29].

### 2.1. Sample and Procedure

The sample design was non-probabilistic for convenience, accessibility, and proximity to the sample. The sample consisted of 451 university students between 18 and 24 years of age from different undergraduate degree programs at the Universidad de La Frontera, Temuco, Chile.

The procedure consisted of inviting the students to participate in the study via email, presenting the research aims and the link to the self-report questionnaire on the QuestionPro platform, which included a form for the respondents to provide their informed consent to participate.

This study has been approved by the Bioethics Committee of the University of Perugia, Italy (protocol number: 2019-17R; date: 14 October 2019).

### 2.2. Instruments

The university students answered an online questionnaire during the first semester of 2021 (March–July), which included a sociodemographic questionnaire and the following scales.

*Family Allocentrism–Idiocentrism Scale* (FAIS): Additionally, known as Lay’s familism scale [7], it assesses the proximity among family members (e.g., “I could not stand that something took me away from what I was doing”). This scale integrates 21 items contained in two subscales: allocentrism (15 items) and idiocentrism (6 items). Participants had to indicate their degree of agreement with each statement using a 5-point Likert scale (1 = strongly disagree; 5 = strongly agree). This scale has been validated in Chinese and Italian adolescent populations [30], and in Brazilian and Argentine populations [31,32]. However, to be used on a Chilean population, the scale was translated and adapted from Italian to Chilean Spanish language and reviewed by a panel of experts following back-translation procedures [33].

Interpretation of the FAIS scale scores is based on the fact that a higher mean score indicates greater approval of the values of familism (allocentrism) and a lower total score indicates lower approval of the values of familism (idiocentrism).

*Depression, Anxiety, and Stress Scale* (DASS-21) [34]: It is a self-report scale designed to measure the negative emotional states of depression, anxiety, and stress experienced within the preceding week (e.g., “I could not stand that something took me away from what I was doing”). This scale integrates 21 items contained in three subscales: depression, anxiety, and stress. The three subscales contain seven items each. Participants had to indicate their degree of agreement with each statement using a 4-point Likert scale (0 = does not describe anything that happened to me or felt during the week; 3 = yes, this happened to me a lot, or almost always). This scale was validated in the Chilean population [35,36]. Interpretation of the DASS-21 4-point Likert scale scores is based on the average score of the subscales, which is itself based on the division of the nominal scores of the scale into three classifications. Scores from 0 to 6 are classified as low, scores from 7 to 13 are classified as medium, and those from 14 to 21 are considered high.

### 2.3. Data Analysis

The descriptive analyses were carried out using the statistical program SPSS. Reliability analysis of the internal consistency of the instruments was performed by calculating the omega coefficient [37].

The testing of the hypotheses was carried out using structural equation modeling (SEM) through the Mplus 7.11 program. The parameters of the structural models were estimated using robust weighted least squares (WLSMV). The fit of the model was evaluated considering the Tucker–Lewis index (TLI), the comparative fit index (CFI), and the root-mean-square error of approximation (RMSEA). It should be noted that a model presents an acceptable fit when the TLI and CFI values are greater than 0.90 and when the RMSEA value is less than 0.08 [38].

Gender, living situation, financing of schooling, and occupation were used as control variables of the model, incorporating the direct effect of these variables on depression, anxiety, and stress.

## 3. Results

The sample consisted of 451 university students, whose average age was 20.9 years. A greater proportion were women (61.9%), lived with their families (86.3%), financed their studies through scholarships (75.6%), and did not carry out work activity (60.5%) (Table 1).

In this study, the range of standardized factor loadings on the Lay’s familism scale varied from 0.428 to 0.767 on the allocentrism subscale and from 0.212 to 0.756 on the idiocentrism subscale. Regarding the level of reliability, an omega of 0.89 was reached for the allocentrism subscale and an omega of 0.64 was obtained on the idiocentrism subscale.

Furthermore, the range of standardized factorial loads on the DASS-21 scale varied from 0.804 to 0.899 on the depression subscale, from 0.432 to 0.874 on the anxiety subscale, and from 0.653 to 0.824 on the stress subscale. Regarding reliability, an omega of 0.94 was reached on the depression subscale, an omega of 0.90 on the anxiety subscale, and an omega of 0.89 on the stress subscale.

Table 2 presents the mean scores of the allocentrism and idiocentrism subscales, as well as on the depression, anxiety, and stress subscales.

The structural model presented an acceptable fit with the data (CFI = 0.900; TLI = 0.894; RMSEA = 0.062). As shown in Figure 1, the results of the estimation of the standardized path coefficients of the structural model showed that allocentrism was positively and significantly associated with depression (*γ* = 0.112, *p* = 0.051), anxiety (*γ* = 0.209, *p* < 0.001), and stress (*γ* = 0.212, *p* < 0.001), corroborating H1. Similarly, idiocentrism was negatively and significantly associated with depression (*γ* = −0.392, *p* < 0.001), anxiety (*γ* = −0.368, *p* < 0.001), and stress (*γ* = −0.408, *p* < 0.001), making it possible to accept H2.

On the contrary, when studying the subscales, it was found that allocentrism was correlated positively and significantly (covariance) with idiocentrism (*r* = 0.376, *p* < 0.001). Depression was positively and significantly correlated (covariate) with anxiety (*r* = 0.699, *p* < 0.001) and stress (*r* = 0.802, *p* < 0.001). Additionally, anxiety was positively and significantly correlated (covariate) with stress (*r* = 0.950, *p* < 0.001).

It is worth mentioning that, when controlling for the variables gender, living situation, financing of schooling, and occupation, a statistically significant positive association was only established between gender and depression (*p* = 0.005), anxiety (*p* < 0.001), and stress (*p* < 0.001) (Table 3), with women experiencing greater depression, anxiety, and stress than men.

## 4. Discussion

The findings suggest that family allocentrism [7], depression, anxiety, and stress [34] subscales presented acceptable values of reliability through the omega coefficient [39]. Furthermore, the family idiocentrism subscale [7] presented an internal consistency of less than 0.70, making it possible to accept values less than 0.70 in some circumstances [40].

The results of this study show that allocentrism was positively and significantly associated with idiocentrism, as they are dimensions of the same concept of familism. Additionally, allocentrism was positively and significantly associated with depression, anxiety, and stress. These results are partially consistent with those of previous studies [8]. Lee and Solheim [8] pointed out that familism (allocentrism) could contradict the need for individuation in young people and even generate psychological discomfort. Other studies have found more depressive symptoms with higher levels of familism [2].

In this regard, we propose to consider that familism describes a relational pattern that, from the perspective of family functioning, is consistent with agglutinated or highly cohesive families [41]. This is characterized by excessive fusion among family members, showing diffuse limits and a lack of differentiation between subsystems. Additionally, it includes overprotective behavior toward children, which inhibits the development of the autonomy needed for an appropriate development.

Agglutinated or amalgamated families tend to be overloaded with internal and external stress. These families present scarce individuality and a lack of differentiation; therefore, personal events might be assimilated as their own by each member of the group.

Our findings allow us to suggest that agglutinated or cohesive family interactions at least constitute a risk factor in the manifestation of depressive syndromes, anxiety, and stress in university students in emergency contexts such as the COVID-19 pandemic. Similar results were presented by Achenbach and Edelbrock [26] who stated that parental behaviors can be predictors of internalized behaviors (depressive syndrome and anxiety, among others), especially in the adolescent population and more frequently observed in women more than in men.

On the contrary, relational dynamics that favor autonomy and personal identity, along with the configuration of a personal life project during adolescence and youth, contribute to the development of the individual, facilitating the transition to adulthood. This can explain the findings regarding family idiocentrism and the negative and significant association to depression, anxiety, and stress.

It is worth mentioning that, when controlling our model for the variables gender, living situation, and method of financing schooling and occupation, a statistically significant positive association was only established between gender and depression, anxiety, and stress, with women experiencing greater depression, anxiety, and stress than men, as reported by other studies [23,24,25,26].

However, the differences in gender in this study can be explained by cultural differences. Previous studies have shown that cultural aspects influence men and women differently [15,18], especially during upbringing and during the development of social relationships.

This study contributes to the existing literature because it demonstrates the importance of promoting behaviors associated with idiocentrism, such as autonomy and individuation, as a way of reducing negative emotional symptoms. Additionally, it allows us to suggest the implementation of promotional activities with university adolescents and with families that help them to achieve greater general well-being.

However, we cannot ignore that this study was carried out during a health emergency due to COVID-19. This emergency involved a modification to people’s lives, especially to the usual relational patterns in the family. For example, some university students did not live with their families during the academic year, while others had to return to their homes but remain in confinement. Chile was considered one of the strictest countries with confinement measures to face the pandemic. During the pandemic, families played a central role in students’ daily lives [42], but it also meant that many families were locked up in a house sharing more time together than usual. This situation could have influenced the results.

These findings are important and should be considered by health and education professionals working with youth and their families. These findings can contribute to supporting actions to reduce negative symptomatology and promote greater well-being in university students.

The following limitations have been identified in this study. First, the cross-sectional design cannot establish causal relations; therefore, only an association of the variables can be made. Second, the non-probabilistic sample and the convenience sampling method did not allow the extrapolation of the results to all Chilean university students. Finally, as the study data were collected by self-reports, the responses obtained through the instrument could have been influenced by social desirability.

It is suggested for future research to consider that adherence to the values of familism might change according to age [13]. It would interest to conduct longitudinal studies and regression analysis to identify life-stage changes and causal relations.

## 5. Conclusions

Our study contributes to the existing literature, demonstrating the importance of promoting behaviors associated with idiocentrism, such as autonomy and individuation, in the university stage, as a way of reducing negative emotional symptoms, promoting independence and the transition to adulthood.

Additionally, the information provided can be used to support preventive and focused mental health actions that contribute to a greater general well-being of this segment of the population.

## Figures and Tables

**Figure 1 ijerph-20-04149-f001:**
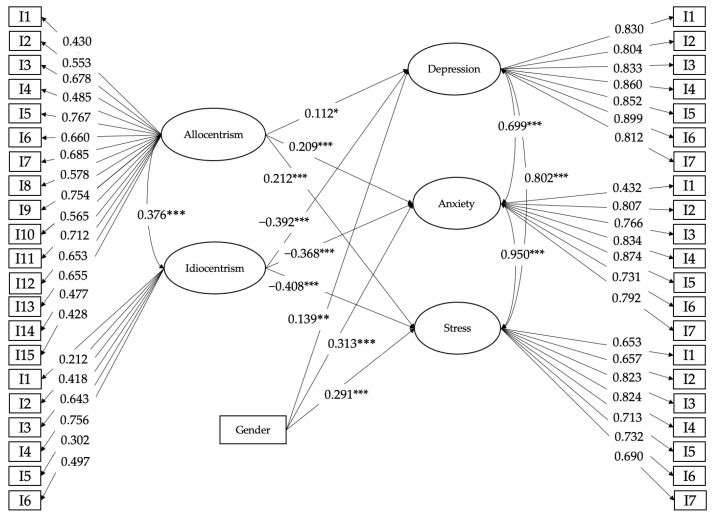
Model of theoretical associations between allocentrism and idiocentrism with depression, anxiety, and stress, together with gender control variables. For greater simplicity of the model, other non-significant control variables were not plotted. * *p* ≤ 0.05, ** *p* ≤ 0.01, and *** *p* ≤ 0.001.

**Table 1 ijerph-20-04149-t001:** Characteristics of the sample (*n* = 451).

Characteristics	Total Sample
Age (mean (SD))	20.93 (3.0)
Gender (%)	
Women	61.9
Men	38.1
Living situation (%)	
Family	86.3
Alone	9.1
Other	4.7
Financing schooling (%)	
Scholarship	75.6
Family	14.9
Itself	1.3
Other	8.2
Occupation (%)	
No	60.5
Full time	7.1
Part time	7.1
Occasionally	23.9
Other	1.3

**Table 2 ijerph-20-04149-t002:** Descriptive statistics.

Scale	Mean	Standard Deviation
1. Allocentrism	48.79	10.26
2. Idiocentrism	17.49	3.98
3. Depression	10.92	6.07
4. Anxiety	10.20	6.00
5. Stress	10.72	5.38

**Table 3 ijerph-20-04149-t003:** Standardized estimates of the effects of control variables.

Control Variable → Dependent Variable	Estimation	*p*-Value
Gender → Depression	0.139	0.005 **
Gender → Anxiety	0.313	0.001 ***
Gender → Stress	0.291	0.001 ***
Living situation → Depression	0.002	0.974
Living situation → Anxiety	0.066	0.145
Living situation → Stress	0.031	0.458
Financing of schooling → Depression	−0.030	0.549
Financing of schooling → Anxiety	−0.016	0.743
Financing of schooling → Stress	−0.042	0.429
Occupation → Depression	0.014	0.783
Occupation → Anxiety	0.082	0.083
Occupation → Stress	0.060	0.222

** *p* ≤ 0.01, and *** *p* ≤ 0.001.

## Data Availability

The data presented in this study are available upon request to the corresponding author. The data are not publicly available for privacy or ethics reasons.

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
