# Peer review of "Association between Familism and Mental Health in College Adolescents during the COVID-19 Pandemic"

_ijerph, 2023, doi:10.3390/ijerph20054149_

Round 1

Reviewer 1 Report

The paper is generally well written and structured. In my opinion the sample and estructural model of analysis highlight the accomplishments in both the social sciences and mental health policies.

Author Response

Thank you for your time reviewing our article.

We really appreciate your comments.

Kind regards,

The authors.

Reviewer 2 Report

This study presents a topic of research interest. A good theoretical and conceptual foundation is presented regarding the value field of allocentrism in the studied population. A suitable design is observed. It is suggested to improve the research proposal in terms of its cross-sectional design and deepen the scope due to the sampling used. Self-report measures are complex to situate without addressing the social desirability factor. It is suggested to improve and expand this section, in order to correct and develop coherent lines of work for future studies. A good delimitation and test of working hypotheses is observed, with appropriate statistical analysis.

Author Response

Reviewer 2

Thank you for your time reviewing our article.

We really appreciate your comments.

Regarding your suggestions, we would like to let you know the following:

  • English language and style have been reviewed and modified.
  • We tried to make the introduction and conclusion clearer.
  • A more descriptive explanation has been written in methods
  • We included the social desirability factor as a limitation and future research opportunities.

    Kind regards,

    The authors.

Reviewer 3 Report

Thank you for the opportunity to review this manuscript. The authors endeavor to explore the relationship between familism and mental health amongst a population of college students during the pandemic. Despite the importance of understanding the influence of this cultural variable on mental health, this reviewer has several reservations with how the study was conducted and how the findings were interpreted. Below are some specific concerns.

Intro/Procedures

1.     The introduction is long and lacks clarity – it would be helpful to have a summary of what we know so far about familism and why we should be interested in understanding the impact of this cultural variable on mental health (particularly during the pandemic). Importantly, the authors do not provide a basis for their hypothesis.

2.     The scale referenced does not appear to match scale used.  It’s unclear why one of many other familism scales were not considered. To what extent does this tap into the construct of familism vs. something else – the sample items presented do not map on to one of the several proposed dimensions of familism.  Also, the items presented for the DASS are the same as those presented for the proposed familism measure.

3.     I had difficulty finding many of the articles cited; please check citations.

Results/Discussions

1.     Please reflect on your finding related to allocentrism being positively correlated with idiocentrism.

2.     Did the authors explore if gender was related to allocentrism?  It’s unclear as written if the authors are trying to explain that after controlling for gender, the relationship between allocentrism and distress changed/did not change. This can impact their interpretation of findings, and it was unaddressed in the discussion section.

3.     In light of concern 2, the authors give (perhaps) too much weight to their findings and interpret beyond what their data allows. As they acknowledge in their introduction, there are many dimensions of familism.  It’s hard to say if their findings likely reflect other factors that could be contributing to distress in this population (that may also be related to familism).  For instance, it’s unclear how to truly interpret the non-significant relationship between living situation and distress.

Author Response

Reviewer 3

Thank you for your time reviewing our article.

And thank you very much for your comments and suggestions. They are undoubtedly a great contribution to our study and we receive them with great gratitude.

Regarding your suggestions, we would like to kindly let you know the following:

  • English language and style have been reviewed and modified.
  • We tried to make the introduction and conclusion clearer. First, improving the English language. Second, re-writing the long paragraphs. And Third, synthetizing the ideas.
  • A more descriptive explanation has been written in methods
  • We included the social desirability factor as a limitation and future research opportunities in the manuscript.
  • Indeed, the items presented for the DASS are the same as those presented for the proposed familism measure. This does not depend on us, but it is how it is.
  • The citations and references have been checked.
  • To answer to your question: why one of many other familism scales were not considered? We can say that the FAIS is the scale most used in the field, showing appropriate test-retest reliability and internal consistency in different cultural population such as China, Brazil, Italy, Argentina, and now Chile (Li et al., 2018; Mafioletti et al., 2010; Omar & Uribe-Delgado, 2011; Sato, 2007). Also, Sato (2007) found that FAIS had the highest internal consistency alpha when compared to those of the Self Construal Scale and the Scale of Horizontal and Vertical Collectivism and Individualism. That is why we chose this Scale.
  • It was included in discussions the finding related to allocentrism being positively correlated with idiocentrism.
  • The rest of your concerns regarding results and discussion we tried to addressed them the best we could.

Li, J.B.; Delvecchio, E.; Lis, A.; Mazzeschi, C. Family allocentrism and its relation to adjustment among Chinese and Italian adolescents. Psychiatry Res. 2018, 270, 954–960. https://doi.org/10.1016/j.psychres.2018.03.036.

Mafioletti Macarini, S.; Dal Forno Marins, G.; Azevedo Reis Sachetti, V.; Vieira Mauro, L. Etnoteorias parentais: Um estudo com mães residentes no interior e na capital de Santa Catarina. Psicol. Reflexão E Crítica 2010, 23, 37–45. https://doi.org/10.1590/S0102-79722010000100006.

Omar, A.G.; Uribe-Delgado, H. Links of allocentrism-idiocentrism with perceptions of justice at work. Rev. Interam. De Psicol. De Psicol. Ocup. 2011, 30, 5–20. Available online: http://cincelcentrodeinvestigacion.org/Revistas/suscripcion/rev30_1/capitulo1_

Sato, T. The family allocentrism-idiocentrism scale: Convergent validity and construct exploration. Individual Differences Research. 2007, 5, 194-200.

Wishing you the best and once again thanking you,

Please receive our best regards.

The authors.